# Towards Generating Realistic Wrist Pulse Signals Using Enhanced One Dimensional Wasserstein GAN

**DOI:** 10.3390/s23031450

**Published:** 2023-01-28

**Authors:** Jiaxing Chang, Fei Hu, Huaxing Xu, Xiaobo Mao, Yuping Zhao, Luqi Huang

**Affiliations:** 1School of Electrical and Information Engineering, Zhengzhou University, Zhengzhou 450001, China; 2Research Center for Intelligent Science and Engineering Technology of TCM, China Academy of Chinese Medical Sciences, Beijing 100000, China

**Keywords:** wrist pulse signal, data augmentation, generative adversarial network

## Abstract

For the past several years, there has been an increasing focus on deep learning methods applied into computational pulse diagnosis. However, one factor restraining its development lies in the small wrist pulse dataset, due to privacy risks or lengthy experiments cost. In this study, for the first time, we address the challenging by presenting a novel one-dimension generative adversarial networks (GAN) for generating wrist pulse signals, which manages to learn a mapping strategy from a random noise space to the original wrist pulse data distribution automatically. Concretely, Wasserstein GAN with gradient penalty (WGAN-GP) is employed to alleviate the mode collapse problem of vanilla GANs, which could be able to further enhance the performance of the generated pulse data. We compared our proposed model performance with several typical GAN models, including vanilla GAN, deep convolutional GAN (DCGAN) and Wasserstein GAN (WGAN). To verify the feasibility of the proposed algorithm, we trained our model with a dataset of real recorded wrist pulse signals. In conducted experiments, qualitative visual inspection and several quantitative metrics, such as maximum mean deviation (MMD), sliced Wasserstein distance (SWD) and percent root mean square difference (PRD), are examined to measure performance comprehensively. Overall, WGAN-GP achieves the best performance and quantitative results show that the above three metrics can be as low as 0.2325, 0.0112 and 5.8748, respectively. The positive results support that generating wrist pulse data from a small ground truth is possible. Consequently, our proposed WGAN-GP model offers a potential innovative solution to address data scarcity challenge for researchers working with computational pulse diagnosis, which are expected to improve the performance of pulse diagnosis algorithms in the future.

## 1. Introduction

Regarded as a traveling pressure generated by the cardiac cycle, the wrist pulse contains a wealth of information on the cardiovascular (CV) system [1]. Poor health condition may cause variations in arterial properties, affecting pulse-related parameters including morphology, strength, rhythm, etc. [2,3]. Pulse diagnosis (PD), which evaluates health status by analyzing tactile radial arterial palpation with trained fingertips, plays an important role in oriental medicine including traditional Chinese medicine (TCM) and traditional Korean medicine (TKM) [4]. In practice, however, its effectiveness is limited by its reliance on long-term training and rich experience, which can lead to significant differences in diagnostic results among different doctors [5,6].

To address these issues, there has been an increase in interest in developing sensors to acquire wrist pulse signals and exploring pattern recognition/machine learning techniques to analyze health conditions [6,7,8,9,10,11], also known as computational pulse diagnosis [12]. Generally speaking, excluding preprocessing, computational pulse diagnosis mainly consists of two parts: feature extraction and pattern classification. For instance, lots of features in time domain or frequency domain (such as fast Fourier transform [13], short Time Fourier transform, Hilbert–Huang transformation [14], Wavelet transform [15,16]), linear or nonlinear (such as multi-scale entropy, Lyapunov exponent) are derived [17]. Regarding pattern classification, similarity-measure-based methods (such as edit distance with the real penalty (ERP) [18]) and conventional machine learning methods (such as support vector machine (SVM), K-nearest neighbor classifier (KNN) and gradient boosting decision trees (GBDT) [19]) were applied in pulse recognition. These above features and classification methods have achieved significant progress, promoting pulse modernization. However, in the above-mentioned methods, feature extraction and pattern classification are mutually independent. Usually, it necessitates an extensive engineering experience from professionals, which is time-consuming and tedious. Additionally, traditional shallow models can hardly represent the complex mapping relationship among pulse signals and the state of human body.

In recent years, deep learning has been revolutionary in computer vision due to its strong modeling and representational ability. With the advantages of its adaptive feature learning capacity and multi-layer nonlinear mapping ability, deep learning models frequently outperform classic shallow methods as an end-to-end process [20]. For wrist pulse recognition, recently a series of work on deep learning has also been employed [5]. To classify pulse signals without extracting complicated features, a nine-layer deep convolutional neural network was proposed [21]. Combining non-threshold recurrence plot (RP) and deep VGG-16 nerwork, Yan et al. proposed a pulse recognition strategy, which outperforms other approaches [22]. Treating the acquired correlations between various features as nodes on a graph, Zhang [23] et al. utilized the graph convolutional network (GCN) to classify pulse signals. A four-layer multi-task fusion convolutional neural network (CNN) for type 2 diabetes detection was constructed by encoding pulse signals into 2D images using several time-series imgaing methods, including the gramian angular field (GAF), Markov transition field (MTF) and recurrence plots (RPs) [24]. Jiang et al. [8] adopted a recurrent neural network (RNN) and long short-term memory (LSTM) as classifiers to identify multiple diseases. In corresponding literature, compared to other conventional approaches, all these mentioned deep learning-based algorithms obtained superior performance.

Compared with traditional pulse diagnosis methods, deep learning methods circumvent feature extraction procedures, extracting the hierarchy latent features in the pulse waveform signal. However, unfortunately, largely owing to (i) data privacy and (ii) time exhaustion when collecting data, the used datasets in existing studies of pulse diagnosis are small (usually hundreds [5,21,23,24]). The paucity of pulse signal collected limits the capabilities of achieving higher performance in deep learning-based models, which makes them incapable of training a robust model. When a diagnosis is much more common than others due to the scarcity of aberrant instances, such performance may be further hampered. Simply put, developing deep learning models with small datasets risks overfitting, greatly limiting generalizability [20].

In computer vision, this major limitation of small dataset is commonly treated by data augmentation methods to increase accuracy, stability and reduce over-fitting. Most notably, since its inception [25], generative adversarial networks (GANs) offer an very competitive alternative by generating artificial data with the same distribution as the original training data from the random noise, thus expanding the training dataset. Basically, GAN consists of two separate neural networks: a generator and a discriminator. During training, the generator captures the real distribution of the original samples to generate new samples, and the discriminator identifies whether the inputs are from the original samples or the generated new samples. Hopefully, the generated new samples are different from the original samples but have a similar distribution, resulting in an expansion of the sample set. To date, GANs have made significant progress in image-based applications [26] and also start being applied to time series data [27,28]. In our closely related biomedical community, for one-dimensional signals, GANs also have begun to show their power for data augmentation, such as electrocardiogram (ECG) [29,30,31], electroencephalographic (EEG) [32,33,34,35] and photoplethysmogram (PPG) [36,37,38]. However, to date, there has not yet been an attempt at exploring the use of GANs for generating wrist pulse signals, to the best of our knowledge. Therefore, its performance remains unclear in the case of wrist pulse signals.

Motivated by these successful applications and also aiming to overcome the data scarcity problem, the present study presents a novel one-dimension generative adversarial networks (GAN) for generating synthetic wrist pulse signals using a small dataset of real pulse signals. We structure this work as follows. In Section 2, we elaborately describe theoretical background about generative adversarial networks, the proposed WGAN-GP network and the evaluation metrics. The database and experimental settings are then introduced in Section 3, followed by a description and discussion of the experimental results and findings in Section 4. Section 5 concludes with conclusions and recommendations for future work. Section 6 summarizes the results of the current work.

## 2. Methods

### 2.1. Vanilla GANs

As shown in Figure 1, the vanilla GANs consist of a generator (abbreviated as *G*) and a discriminator (abbreviated as *D*). $\theta_{G}$ and $\theta_{D}$ represent the generator and discriminator parameters, respectively. Moreover, $x_{r}$ is sampled from the real data and $x_{f}$ from the simulated data. The input of the generator is denoted as *z*, which is usually sampled from uniform or Gaussian noise distribution. G(z) represents the generated samples by *G*, which is expected to have data distribution similar with original samples. Then, the generated samples are fed into the discriminator to assess their similarity with the real data, which also improve continuously as well.

More specifically, the training between the generator *G* and the discriminator *D* can be formulated as a two-player minimax game
(1)$$\min_{\theta_{G}}\max_{\theta_{D}}V\left(D,G\right)=E_{x_{r}∼P_{r}}\left[\log\left(D\left(x_{r}\right)\right)\right]+E_{z∼P_{z}}\left[\log\left(1-D\left(G\left(z\right)\right)\right)\right]$$
where $P_{r}$ is the real data distribution and $P_{z}$ is the random noise distribution, D(x) is the probability that *x* is derived from real data.

### 2.2. Proposed Wasserstein GAN with Gradient Penalty Term

An important drawback of vanilla GAN in practice is the infamous volatility of the discriminator during training. If the discriminator collapses during training and only recognizes a small number of narrow modes of the input distribution as real, the generator will only create a limited range of outputs. The recent years have seen significant improvements in sample quality and training stability. Among others, it has been demonstrated that the WGAN-GP loss function, which uses a gradient penalty, efficiently increases training stability and convergence [39].

Wasserstein GAN (WGAN) [40] is the foundation for WGAN-GP. In order to discriminate between real and simulated data distributions forWGAN, the so-called earth moving distance, also known as the Wasserstein distance, is used.
(2)$$\bar{W}\left(P_{r},P_{f}\right)=E_{x_{r}∼P_{r}}\left[D_{\theta_{D}}\left(x_{r}\right)\right]-E_{x_{f}∼P_{f}}\left[D_{\theta_{D}}\left(x_{f}\right)\right]$$
The discriminator is trained to minimize the Wasserstein distance for a fixed generator or $\theta_{G}^{*}$,
(3)$$L_{D}\left(\theta_{D},\theta_{G}^{*}\right)=\bar{W}\left(P_{r},P_{f}\right)$$
$P_{f}$ stands for the simulated data distribution. The training for the generator entails maximizing the following loss with a fixed $\theta_{D}^{*}$.
(4)$$L_{G}\left(\theta_{G},\theta_{D}^{*}\right)=E_{x_{f}∼P_{f}}\left[D_{\theta_{D}^{*}}\left(x_{f}\right)\right]=E_{x_{f}∼P_{f}}\left[D_{\theta_{D}}\left(G_{\theta_{G}^{*}}\left(z\right)\right)\right]$$

Since *D* requires to be K-Lipschitz in WGAN, the initial response was to limit the discriminator’s weights to the range [−c,c] in order to satisfy these requirements. Even though stability has greatly improved, using weight clipping directly to impose a Lipschitz constraint can occasionally result in subpar samples or a failure to converge [39]. As an alternative, WGAN-GP implemented a solution by including the following gradient penalty to the WGAN loss in order to more effectively impose the Lipschitz continuity on the discriminator.
(5)$$\tilde{W}\left(P_{r},P_{f}\right)=\bar{W}\left(P_{r},P_{f}\right)+\lambda E_{\hat{x}∼P_{\hat{x}}}[(∥\nabla_{\hat{x}}D\left(\hat{x}\right){∥}_{2}-1{)}^{2}]$$
where $\hat{x}$ denotes the samples on the line between $P_{r}$ and $P_{f}$ and λ is a hyper-parameter that modifies the trade-off between the WGAN loss and gradient penalty.

### 2.3. Architecture Details

According to previous work on GAN, the structure configuration and training setting of GAN are particularly crucial for model optimization. For configuration, we followed the the model architecture suggested in [39], which is illustrated in Table 1. Basically, we designed our networks according to the setup described in Karras et al. [41]. They showed that by gradually raising the resolution of the network, the quality of the generated sampless could be improved. Firstly for the first layer of network input, we empirically select 100 sample points. Accordingly, to reach the final resolution of 2000 samples, we gradually raise the resolution by a factor of 2, that is, over five steps. Also note that, for WGAN, batch normalization is frequently used in both the generator and the discriminator to help stabilize training. However, for the discriminator in WGAN-GP, layer normalization is instead used due to that WGAN-GP penalizes the norm of the discriminator’s gradient with respect to each input independently, and not the entire batch [39].

More precisely, it is divided into two components:**Generator:** The generator component takes an *N*-dimensional noise vector (in our case N=100). Then it is feed to the first block consists of a 1D transposed convolutional layer with 1×125 kernel size. It has four hidden layers; each block consists of with a 1D transposed convolutional layer with 1×4 kernel size, then a batch normalization followed by ReLU activations. Note that Tanh activations are employed for the last block. Finally, the generated signal of the output serves as input to the subsequent discriminator component.**Discriminator:** The discriminator component is learned in a supervised manner to minimize the error for classifying false signals from real signal samples. Specifically, it has three hidden fully connected layers. Each block consists of a 1D convolutional layer with 1×4 kernel size, then a instance normalization followed by Leaky ReLU activations. The last block consists of a 1D transposed convolutional layer with 1×125 kernel size. Finally, the output layer computes the probabilities of the two classification categories: simulated or real.

## 3. Materials and Experiment Design

### 3.1. Data Collected

The study recruited 320 healthy college students (166 females and 154 males), summarized in Table 2. The study was fully disclosed to all participants, who gave their informed consent. After a 10-minute rest period during which personal information (age, height and weight) was acquired and the study protocol was given. All measurements were performed in a quiet environment and pulse collectors had been trained by professional doctors. The Chun position of left wrist pulse was chosen and recorded for 10 s for each subject using pressure sensor: ZM-300 intelligent TCM pulse pressure detector (manufactured by Shanghai University of Traditional Chinese Medicine in Shanghai, China, as shown in Figure 2), which is mainly composed of single-head pulse transducer, pulse amplifier, A/D conversion card and computer, among which the pulse amplifier is composed of two parts: pulse collector AC amplification loop (pulse wave loop) and DC amplification loop (pulse pressure loop), similar to that of TCM in perceiving the pulse. The acquired pulse signals were sampled at 200 Hz. That gives a total of 2000 samples for each subject.

### 3.2. Preprocessing

As a weak physiological signal, acquired pulse data are easily interfered by other signals, including the subject’s limb and power frequency interference [5]. For the preprocessing of the pulse signals, we adopted the denoising and baseline drift correction methods in [42]. Specifically, the effective frequency of the pulse signal is generally 0∼10 Hz and will not exceed 40 Hz. First, a low-pass filter with a cut-off frequency of 40 Hz was adopted to filter the original pulse signal. As for the baseline drift, the wavelet-based cascaded adaptive filter method proposed in [43] was employed. Finally, the obtained pulse data points were min-max normalized to be within [0,1].

### 3.3. Model Training

For adversarial training in WGAN-GP, the discriminator and generator are not trained equally, but the discriminator is trained until optimality first. In one epoch, we trained the discriminator for five iterations, as originally proposed by Arjovsky et al. [41], then the generator for one iteration. The gradient clipping value in the loss is set 0.01 and the λ is set 10, as originally recommended by Gulrajani et al. (2017) [39]. The optimizer used in training is the ADAM optimizer with lr=0.0001, $\beta_{1}=0$ and $\beta_{2}=0.9$. For the generator model, as mentioned, the input latent variables *z* size is empirically sampled from 100 dimension normal distribution N(0,1).

Since there is no standardized architecture agreed upon in the literature that applied into generated wrist pulse signals, we performed an in-depth study by comparing with other typical families of GANs, i.e., (1) vanilla GAN [25], (2) DCGAN [44] and (3) WGAN [40]. Specifically, for DCGAN/WGAN, we refer to the original architecture in which Binary CrossEntropy serves as the loss function and also batch normalization were empoyed for both generator and discriminator. For WGAN, only the loss function is different from WGAN-GP and we use the exactly same architectures as WGAN-GP. In addition, the training of the all these GAN models followed the same protocol and we use the PyTorch framework to implement the experiments. For each network, the model was trained for 3000 epochs in batches of 32 examples.

### 3.4. Performance Evaluations for the Generated Samples

While accurately and fairly evaluating different generative models remains a challenge, there are certain reasonable and widely accepted measures [45,46]. In general, a well-trained GAN can implicitly learn the distribution of the original fully observed dataset. Following similar guiding principles, we seek out commonly adopted methods or metrics, which are mainly divided into qualitative visual inspection and quantitative metric evaluations.

#### 3.4.1. Qualitative Visual Inspection

Visual inspection of synthetic pulse data is considered as the intuitive way and also quite often to inspect GANs based models [45]. In this respect, first, time samples and frequency spectrum of generated pulse are employed to show how visually similar a generated pulse was to the real pulse. Moreover, auto-correlation plots (ACF) [47,48], showing the similarity between observations as a function of time lags between them, are also adopted. The definition of autocorrelation $ACF_{k}$, with *k* being the time lag, is given in Equation (Equation 6).
(6)$$ACF_{k}=\frac{\sum_{t=1}^{T-k}\left(x\left(t\right)-\bar{x}\right)\left(x\left(t-k\right)-\bar{x}\right)}{\sum_{t=1}^{T}{\left(x\left(t\right)-\bar{x}\right)}^{2}}$$
where *T* is the length of the pulse signal and $\bar{x}$ is the mean. This helps to compare the consistency of the generated samples with the groud-truth in terms of the long-term temporal trend.

#### 3.4.2. Quantitative Metric Evaluations

Following similar approach [32], three quantitative indicators: maximum mean deviation (MMD) [49], sliced Wasserstein distance (SWD) [50] and percent root mean square difference (PRD) [51], were used to evaluate GANs, which are commonly adopted to how well the generated distributions resemble the original distributions. In general, the smaller these metrics, the closer the distributions are to each other.

**MMD:** The dissimilarity of two probability distributions is assessed by MMD using samples that were taken independently of each distribution [50]. In practice, we calculate the square of the MMD defined below. We use Gaussian RBF kernel $K\left(x,x^{\prime }\right)=\sum_{j=1}^{k}e^{-\alpha_{j}\left.|x-x^{\prime }\right.{|}^{2}}$, where the bandwidth α is equal to the pairwise distance between the joint data. The similarity between the distributions is inversely correlated with the MMD statistic.
(7)$$\begin{array}{cc} {\mathrm{MMD}}^{2}= & \frac{1}{M(M-1)}\sum_{i\neq j}^{M}K\left(x_{i},x_{j}\right)-\frac{2}{M^{2}}\sum_{i,j=1}^{M,M}K\left(x_{i},{\hat{x}}_{j}\right)\\ \; & +\frac{1}{M\left(M-1\right)}\sum_{i\neq j}^{M}K\left({\hat{x}}_{i},{\hat{x}}_{j}\right) \end{array}$$
where $x_{i}$ are $i^{th}$ real samples and the corresponding generated samples are denototed as ${\hat{x}}_{i}$, *M* is the number of total pulse samples.

**SWD:** The Wasserstein distance expresses the price of changing one distribution into another given a cost function [39]. The sliced Wasserstein distance is a 1d projection-based approximation of the Wasserstein distance. By computing the Wasserstein distance between each one dimensional (slice) projection, it approximates the two Wasserstein distance distributions. This approach offers the benefit of a closed solution and an associated quick computation for one-dimensional situations. In practice, a limited set of random one-dimensional projections can approximate the Wasserstein distance of the slice itself. For more details, please refer to [50]. The lower slice Wasserstein distance shows that the sample variation and appearance of the two distributions are similar.

**PRD:** Another popular distortion measurement technique at the moment is PRD, which measures the difference between real data and created points. The disparity between the original signal and the reconstructed signal is quantified.
(8)$$\mathrm{PRD}=\sqrt{\frac{\sum_{t=1}^{T}{\left(x\left(t\right)-\hat{x}\left(t\right)\right)}^{2}}{\sum_{t=1}^{T}x^{2}\left(t\right)}\times 100}$$
where x(t) are real samples and $\hat{x}\left(t\right)$ are generated samples. The generated data contains less distortion than the original data with a lower RRD.

## 4. Results

### 4.1. Visual Inspection

#### 4.1.1. Time Samples

The time examples of the ground truth and generated data using different GAN models are displayed in Figure 3. On the whole, all models have a general trend with the original signal. Among them, evidently GAN performed the worst, containing more noise-like data. DCGAN performed poorly with obvious deviation. Given that its primary goal of network configuration was generating images, it is possible that it was not able to capture well almost of the statistical features of wrist pulse signals that were evaluated. WGAN works well, but the details are still not good enough. Conversely, WGAN-GP reached the best approximation of the real data without losing too much detail.

#### 4.1.2. Frequency Spectra

Similarly, Figure 4 shows spectral distribution of artificial signals generated by the four architectures compared to the real signals. We calculated the power spectral density (PSD) of each signal using Welch’s periodogram with a Hanning window of 256 points, 128 points overlap and 1024 points for the fast Fourier transform. The Welch function in the signal module of Python’s SciPy library was used for calculation.

Generally speaking, it can be seen that, for all models, below 20 Hz (the main frequency range of the pulse signal), the similarity is better. However, in the part of high frequency (above 40 Hz), except WGAN-GP, other models deviate to real signals to a large degree. Below 40 Hz, WGAN and WGAN-GP show a good fit, whereas WGAN-GP overall struggles to approximate frequency trends of the raw signal. Relative to the time domain, the advantage of applied weight clipping in WGAN-GP is more obvious.

#### 4.1.3. Auto-Correlation Plots

In Figure 5, we plot auto-correlation plots (ACF), showing the similarity between observations as a function of time lags. It is apparent from the figure that, again, WGAN-GP performed reasonably best, meaning that it can generate data that have similar statistical characteristics to the original. Also note that, for vanilla GAN or DCGAN models, especially the second peak (about 125 in original data) in the ACF, they all have the wrong second peak, and for WGAN, the peak value is also small. For above 150 lags, the ACF are totally different from the original for the first three GAN models. In some sense, the auto-correlation is related to the similarity of the pulse signal. From its time-domain waveform, the pulse signal is similar to the quasi-periodic signal. The results indicate that there is a large amount of deviation between time cycles in these generated pulse data, except for WGAN-GP, which has quite consistent trends with that of ground truth data.

### 4.2. Quantitative Evaluation

Table 3 summarizes the overall adopted metric results to compare real and synthesized wrist pulse data for different architectures, where the best results are written in bold type. We find that WGAN-GP consistently produces the lowest scores for all metrics, which is a significant improvement compared to the vanilla GAN model. Except for SWD, WGAN-GP holds slightly better than the WGAN model. For other metrics, WGAN-GP overtakes other GAN variants with obvious advantage. DCGAN was clearly the worst performing model, possibly because it is not suitable for 1D signals and is originally intended to generate synthetic images. Compared with the vanilla GAN network, the performance of WGAN has been greatly improved. Overall, WGAN-GP is still the best performing model, well consistent with visual inspection.

### 4.3. Stability of Pulse Signal Generation

For the final experimentation, we evaluate the stability of proposed GAN-GP model during training time. According to the above evaluations, the performace of WGAN and WGAN-GP are better. Therfore, the following compasions are conducted between these two models. Under different epochs, these signals synthesized by the generator network of WGAN or WGAN-GP model are illustrated in Figure 6. Overall, for both GAN models, the quality of the generated data increased with the epoch and the training dynamics of WGAN-GAN are quite stable. In compasion, WGAN-GP is capable of generating pulses, with a quality that is clearly better than the ones produced by WGAN under the same epoch.

For example, in the early stage of training (500 or 1000 epochs), WGAN can only capture several cycles. In contrast, WGAN-GP quickly generates roughly similar pulse data with only a few cycles that are not well modeled. As the training process proceeds, a relatively perfect pulse signal can be generated when the epoch is equal to 1500. From the comparison of the above content, it can be concluded that the WGAN-GP model as a whole has strong stability.

## 5. Discussion

With this work, we presented a first attempt at generating wrist pulse signals. For this purpose, several different GAN architectures were implemented and compared with both objective and subjective evaluations. To summarize, the main contributions of this paper can be highlighted as follows:We adapted a novel, specifically designed Wasserstein generative adversarial networks with gradient penalty (WGAN-GP), which has been shown to be effective in improving the training stability and convergence [39] in comparison to vanilla GAN. Furthermore, most of the GAN models are currently optimized for 2D images, which is difficult for direct application in time domain signals. In the present study, one-dimensional convolutional neural network (1D-CNN) is adopted as the building block of both generator and discriminator and carefully tailored to leverage its ability to learn local and hierarchical representations from raw data, thereby allowing the adaptation of the GAN framework to pulse data.We adopted a set of metrics to quantitatively and visually evaluate the quality of the generated samples comprehensively. For visual comparison, the frequency and time-domain, also including auto-correlation plots, were carefully examined between real and generated pulse signals. In terms of quantitative metrics, three statistical analysis: maximum mean deviation [49], percent root mean square difference [51] and sliced Wasserstein distance [50], were employed.We compared the performance of the proposed method with the existing several popular variants of GAN in literature, including vanilla GAN [25], DCGAN [44] and Wasserstein GAN [40]. The various conducted experimental results using real collected wrist pulse dataset demonstrate the effectiveness and advantage of WGAN-GP, showing that the data generation ability of the proposed framework well reflects the distribution of real data.

Overall, the findings are encouraging. Taking these results together, we found that the proposed WGAN-GP reproduced an excellent data generation, demonstrating that it is possible to generate artificial wrist pulse signals from a small ground truth data.

Although we have shown some great progress for wrist pulse generation, it does not mean that the problem of a universal wrist pulse signal generation method is solved. Given the fact that this research field of pulse generation is still at its early stages, we came across various challenges that have not been discussed specifically in the literature. In the following, we compile a list of problems with our study and also list a set of promising directions for better wrist pulse generation in a broader sense. We hope the community finds this work essential to advance as well.

First, on one hand, although overall the WGAN-GP method provides promising results, there are some issues and limitations in the present study:A major limitation of time series GANs is the restrictions placed on the length of the sequence specified that the architecture can manage. In our study, the length of pulse signal (2000) is relatively long compared to other physiological signals (for example, ECG/PPG/EEG usually several hundreds lengths in [30,31,33,34,36,38]). In practice, the longer signal usually requires a longer training time while worsening the performance of unstable network and requiring larger epochs. For example, we found that the modal collapse emerges after about 700 epochs for DCGAN. The pulse signal consists of different cycles, and the length of a cycle is roughly 150. Therefore, decomposing pulse into different cycles and training GANs to generate a single cycle signal may be a possible solution.On the other hand, to balance the training cost, the depth of the generator network we used is relatively shallow. As mentioned earlier, we started with a resolution of 100 time samples and doubled it in five steps to reach 2000 samples. Several design choices, such as convolution size, may also impact the performance. In the future, increasing the depth of the network may better capture the pulse characteristics.In the present study, WP-GAN was trained independently of the classification task. In practice, the possibility to generate wrist signals with associated labels, especially in the scarcity of abnormal cases, is worthy of further study. That is, whether the computational pulse diagnosis performance can be further improved with these generated pulse signals of different categories. For instance, class-conditioned GAN classifier with WGAN-GP can be employed to use category labels as the auxiliary information and therefore allow for generating labeled data [52]. This is what our following work will focus on, with subsequent collection of wrist pulse signals of different diseases.

Second, on the other hand, with the first step for pulse data generation this new field, there are also many open possibilities worth further expansion or investigations in future study to achieve better results:Better network structures: As mentioned, the used wrist pulse signals are relative long (2000 samples), and also the pulse signal is similar to quasi-periodic with each cycle highly correlated. It has been pointed out that relying solely on the convolution operator limits the ability of GAN to capture long-range dependencies across input sequences due to the local receptive field of the convolution operator [53]. To address this problem, self-attention generative adversarial networks (SAGANs) introduce a self-attention mechanism into convolutional GANs, and exhibits a better balance between the ability to model long-range dependencies [53]. In the image [53] and speech domain [54], it has been empirically proved to achieve a superior performance than the baseline. Consequently, introducing a self-attention module into the convolution-based GAN model for pulse data generation may perform better.Better cost function: Another important aspect that requires further investigation is to choose appropriate cost function to improve the generation performance, which is orthogonal to network structures, especially considering that evaluating the quality of the generated samples is still an open research topic [46]. Without knowing what the ultimate goal of the learning process is, the selection of optimization functions is to some extent a random process [55]. Furthermore, the loss function utilized in GANs is important for reducing model training error and speeding up network convergence [55]. For example, a reward term with encouraging intraclass/interclass diversity was tailed to achieve better performance in PPG-GAN models [37]. Using different loss functions with various regularizations may result in a better convergence and further improve data generation performance.Better model training: In comparison to vanilla GAN, WGAN-GP serves as an effective improvement to enhance the training stability and reduce mode collapse. Moreover, a variety of improved variants are also emerging, with the same aim. For example, different solutions such as normalization and regularization schemes [56,57] have been proposed and have demonstrated their superior performance. Consequently, the proposed WP-GAN model has space for further optimization, which could be achieved by adjusting these regularization and normalization techniques.

## 6. Conclusions

At present, there is a growing interest in applications of machine learning into computational pulse diagnosis. However, the small number of existing dataset samples presents an important obstacle to train more powerful deep learning model. In this study, we proposed a novel generation method based on one dimensional GAN as one promising way for wrist pulse data augmentation, which has not been deeply investigated yet. It seems that one-dimensional GAN is an excellent methodology to capture the dynamics of wrist pulse signals. Wasserstein training with gradient penalty consistently outperforms several typical GAN variants.

In summary, our proposed WGAN-GP model offers a potential innovative solution to address data scarcity challenge, for researchers working with computational pulse diagnosis in TCM community. Moreover, since this research field is still at its early stages, there are now many open possibilities for further investigation. For example, introducing a self-attention mechanism with better loss functions or regularization strategies is expected to further improve the performance of generated pulse data.

## Figures and Tables

**Figure 1 sensors-23-01450-f001:**
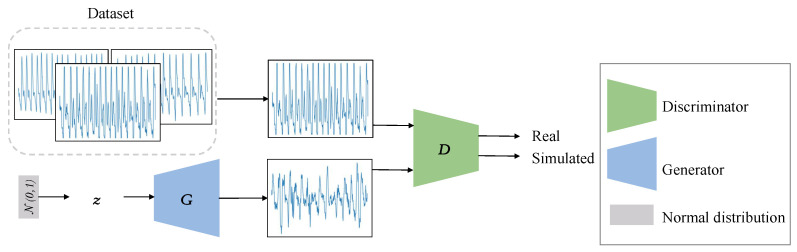
Architecture of the Vanilla GAN.

**Figure 2 sensors-23-01450-f002:**
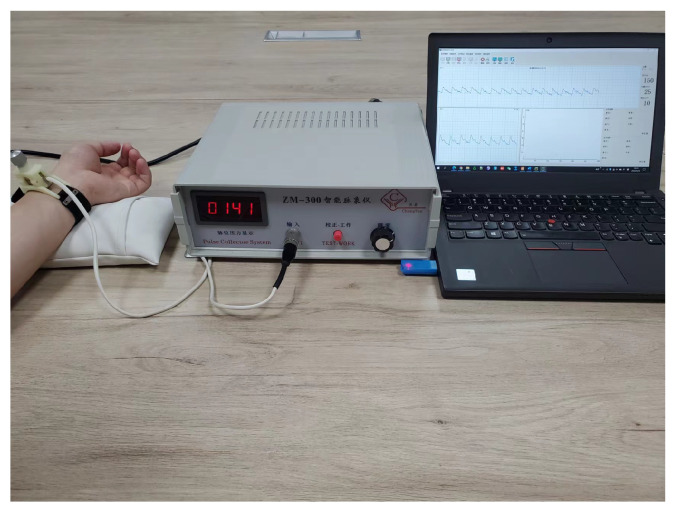
The pulse wave acquisition system.

**Figure 3 sensors-23-01450-f003:**
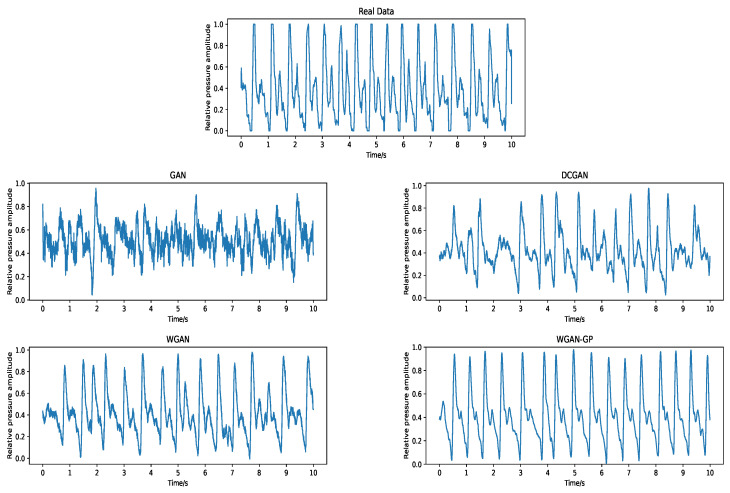
Example of the original and generated signals with different GAN models. In the figure, the real pulse data is shown above, and the false pulse generated by the corresponding GAN model is shown below.

**Figure 4 sensors-23-01450-f004:**
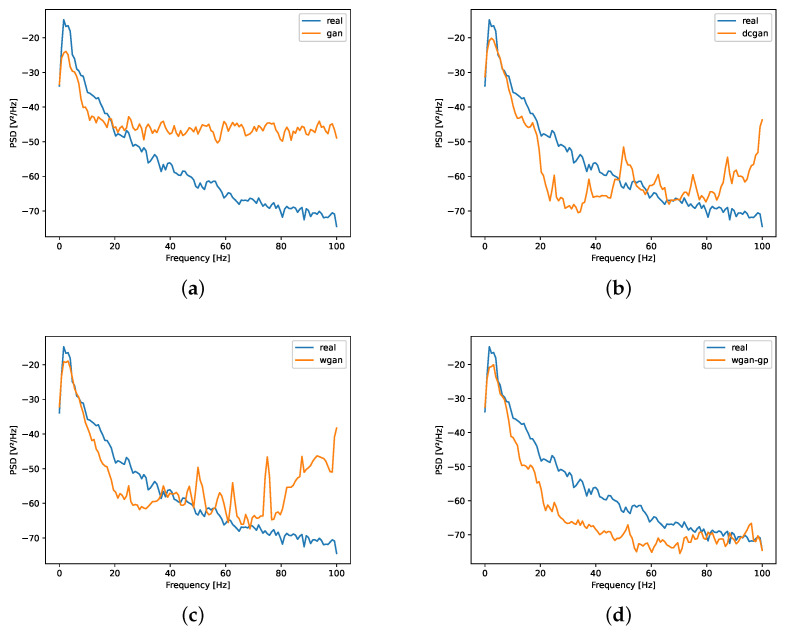
Comparison of the distribution of frequency spectra real signals and signals generated by different GAN architectures: (**a**) GAN; (**b**) DCGAN; (**c**) WGAN; (**d**) WGAN-GP.

**Figure 5 sensors-23-01450-f005:**
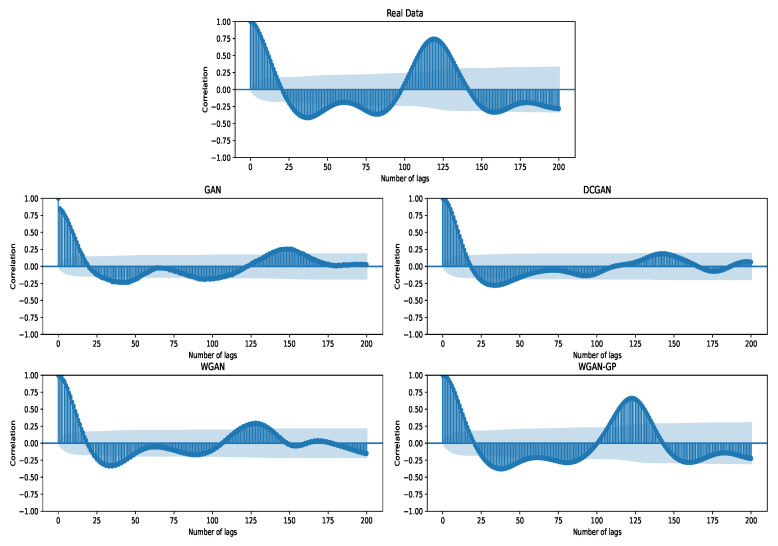
The auto-correlation plots of generated wrist pulse signals with different GAN models. In the figure, the real pulse data are shown above, and the false pulse generated by the corresponding GAN model is shown below.

**Figure 6 sensors-23-01450-f006:**
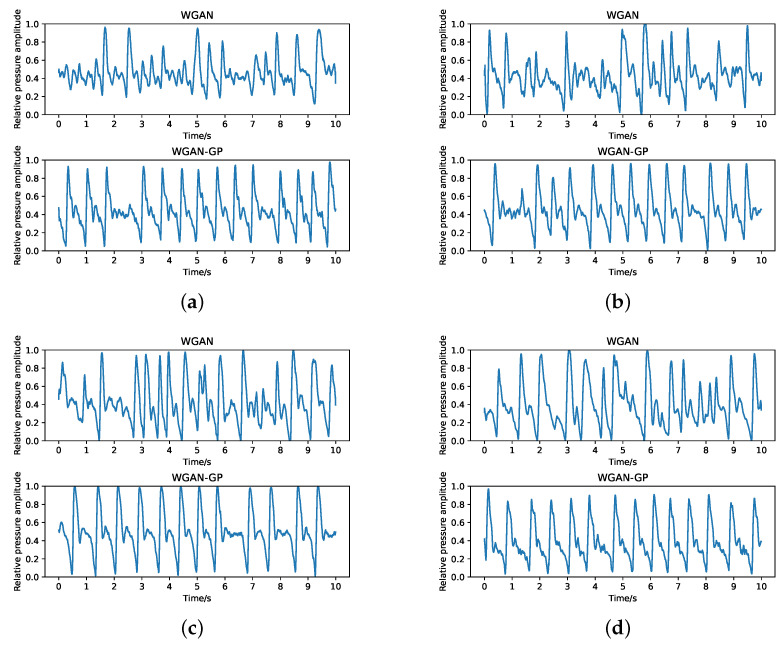
Examples of time series pair generated from WGAN and WGAN-GP models for associated epochs, respectively: (**a**) 500 epochs; (**b**) 1000 epochs; (**c**) 1500 epochs; (**d**) 2000 epochs. In every subfigure, the top is the simulated pulse generated by WGAN and the bottom is the simulated pulse generated by WGAN-GP.

**Table 1 sensors-23-01450-t001:** The adopted architecture of the proposed generator and discriminator in our GAN-based data augmentation for wrist pulse signals.

Generator	
**Layer**	**Output Shape**
Input layer	100 × 1
Transposed Conv1d	521 × 125
ReLU	521 × 125
Batch Norm	521 × 125
Transposed Conv1d	256 × 250
ReLU	256 × 250
Batch Norm	256 × 250
Transposed Conv1d	128 × 500
ReLU	128 × 500
Batch Norm	128 × 500
Transposed Conv1d	64 × 1000
ReLU	64 × 1000
Batch Norm	64 × 1000
Transposed Conv1d	1 × 2000
Tanh	1 × 2000
**Discriminator**	
**Layer**	**Output Shape**
Input layer	1 × 2000
Conv1d	64 × 1000
LeakyReLU	64 × 1000
Instance Norm	64 × 1000
Conv1d	128 × 500
LeakyReLU	128 × 500
Instance Norm	128 × 500
Conv1d	256 × 250
LeakyReLU	256 × 250
Instance Norm	256 × 250
Conv1d	521 × 125
LeakyReLU	521 × 125
Instance Norm	521 × 125
Conv1d	1 × 1

**Table 2 sensors-23-01450-t002:** Total subject characteristics (mean value ± standard deviations).

Variable	Male (154)	Female (166)
Age (year)	20.21 ± 0.53	20.35 ± 0.46
Height (cm)	175.32 ± 4.25	161.35 ± 4.36
Weigth (kg)	65.32 ± 8.35	55.07 ± 5.35

**Table 3 sensors-23-01450-t003:** Comparison of the metric results (mean value ± standard deviations) for different GAN network architectures (lower is better).

Method	MMD	SWD	PRD
GAN	2.4777 ± 0.1393	0.0519 ± 0.0097	8.0296 ± 0.6404
DCGAN	5.1677 ± 0.0864	0.2545 ± 0.0053	11.9029 ± 0.5347
WGAN	0.2815 ± 0.0596	0.0145 ± 0.0043	6.7092 ± 0.5350
WGAN-GP	**0.2325 ± 0.1003**	**0.0112 ± 0.0064**	**5.8748 ± 0.5630**

## Data Availability

Where data is unavailable due to privacy or ethical re-strictions.

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
