# Peer review of "Towards Generating Realistic Wrist Pulse Signals Using Enhanced One Dimensional Wasserstein GAN"

_sensors, 2023, doi:10.3390/s23031450_

Round 1

Reviewer 1 Report

The authors describe a deep learning, WGAN approach to analysing pulse data. The paper is generally well written, clearly explained and an interesting study. However, some revisions are essential.

    Section 3.1. is methods not results and should be moved into the Methods. More details should be given on the methods, ethical information is essential - was the study approved by an ethics review board (I realised later this was listed at the end of the paper - it is useful to include this in the methods as well). Why only collect 10 seconds of data - this seems very short. Filtering 10 seconds of data may also cause spurious effects at the start and end - how was this avoided? Were shorter epochs used for the final analysis? You mention in the discussion that longer lengths can cause issues but given that the frequency of interest is up to 40Hz would it be better to record with a lower sampling rate?   Table 2 - please state how many were male and female. Why does it say n=57 when the methods say that 320 participants were recruited?   Table 3 - presumably these values are mean values across all the data/networks you trained? Please also give some measure of variation e.g. standard deviation
Minor: Line 167 'instead layer normalization serves as recommended a drop-in replacement due to that WGAN-GP penalize the norm  - this sentence doesnt makes sense and should be rephrased.   line 172 'Then it is feed to the first block consists of with’ - this sentence doesnt makes sense and should be rephrased.   line 182 'The last block consists of with a 1D’ - this sentence doesnt makes sense and should be rephrased.   line 284 ‘overbeats’ - this is not a real word, please rephrase.   line 297 'quality clearly inferior’ - I’m not sure this is what you mean - should this say that the WGAN-GP is superior? At the moment what you say implies that the WGAN-GP is not as good as the WGAN.

Reviewer 2 Report

I have read the manuscript. It is an interesting work, but it presents issues that must be solved before publication. Following, some comments:

                The manuscript presents several typos that should be fixed. Moreover, the language is not scientific (e.g., I suggest using the term “simulated” instead of “fake”).

                Quantitative results should be presented in the Abstract.

     I suggest revising the Introduction section. I suggest concluding the Introduction only with the aim of the paper; thus, I suggest removing the innovation/contribution of the paper on the literature, that should be deeply discussed in the Discussion section.

                I suggest revising the Methods section. Specifically, I suggest removing the sentences that introduce the content of each paragraph.

                Materials, Preprocessing and Statistics (Quantitative evaluation) should be placed in the Method section instead of Results section. In the Results section, only quantitative results should be reported.

                The number of subjects in the Table 2 (n=57) and the number of subjects in the text (n=320) are not matching. Moreover, I suggest reporting in Table 2 the cardinality of males and females.

                Quality of Figure 3 is very low. I suggest reporting the time in x-axis (in seconds) and the units of the PPG amplitude in y-axis.

                Quantitative evaluation should be better described. Indeed, the computation of the presented statistical indices is not clear.

                The quantitative comparison with other methods should be reported.

Reviewer 3 Report

General comment:

This manuscript describes a Wassertein GAN with gradient penalty algorithm for generating wrist pulse signals. The work is relevant in the field of data modeling using scarce data coming from health measurements. Furthermore, the applications of current trends in data analytics are mandatory to strengthen its powerful and versatility. The conceptual framework is clear and the results are well supported. The manuscript is interesting and well written. I have some points that should be addressed before the manuscript can be accepted.

Comment 1:

In page 1, line 16, the word “oer” seems wrong, may be “offers”.

Comment 2:

It could be useful to add more technical details about the ZM-300.

Comment 3:

Regarding the frequency-domain analysis of the generated signals. The authors should include how did the spectra is computed. Is there a windowing process? Which is the number of points for the computation? Is there a spectral estimation algorithm?

Comment 4:

It is not clear what motivates the computation of the ACF. Physically which is the meaning of the ACF for the pulse waves?

Comment 5:

In line 277, a “t” is missing at the end of the word “excep”.

Comment 6:

Which is the main advantage of using generative adversarial network-based method to generate the pulse signals, with respect to a hardware-based approach? It should be emphasized in the manuscript.

Round 2

Reviewer 2 Report

I read again the paper. The paper improved, but the language and the format should be revised. I suggest a language check before publication.
